# Immune Checkpoint Inhibitor Therapy for Metastatic Melanoma: What Should We Focus on to Improve the Clinical Outcomes?

**DOI:** 10.3390/ijms251810120

**Published:** 2024-09-20

**Authors:** Sultana Mehbuba Hossain, Kevin Ly, Yih Jian Sung, Antony Braithwaite, Kunyu Li

**Affiliations:** Department of Pathology, Dunedin School of Medicine, University of Otago, Dunedin 9016, New Zealand; mehbuba.hossain@otago.ac.nz (S.M.H.); yihjian.sung@otago.ac.nz (Y.J.S.); antony.braithwaite@otago.ac.nz (A.B.)

**Keywords:** ICI, melanoma, treatment resistance, epigenetic regulation, microbiota, anti-tumour immune response, predicting treatment outcomes

## Abstract

Immune checkpoint inhibitors (ICIs) have transformed cancer treatment by enhancing anti-tumour immune responses, demonstrating significant efficacy in various malignancies, including melanoma. However, over 50% of patients experience limited or no response to ICI therapy. Resistance to ICIs is influenced by a complex interplay of tumour intrinsic and extrinsic factors. This review summarizes current ICIs for melanoma and the factors involved in resistance to the treatment. We also discuss emerging evidence that the microbiota can impact ICI treatment outcomes by modulating tumour biology and anti-tumour immune function. Furthermore, microbiota profiles may offer a non-invasive method for predicting ICI response. Therefore, future research into microbiota manipulation could provide cost-effective strategies to enhance ICI efficacy and improve outcomes for melanoma patients.

## 1. Introduction

Immune checkpoint molecules (ICMs), such as CTLA-4 and PD-1, are key immune checkpoint regulators involving T cell exhaustion and tolerance to prevent excess immune response and pathology [1]. However, tumours often hijack these regulatory mechanisms to evade immune detection [2]. ICIs are monoclonal antibodies designed to block these signalling pathways, thereby countering T cell exhaustion and tolerance, leading to a prolonged anti-tumour immune response [3]. ICIs have significantly transformed the treatment of advanced melanoma, leading to substantial improvements in long-term progression-free survival for many patients [4,5]. These therapies target immune checkpoint molecules (ICMs) such as cytotoxic T-lymphocyte-associated protein 4 (CTLA-4) and programmed cell death protein 1 (PD-1), which are involved in T cell exhaustion and tolerance. By inhibiting these checkpoints, ICIs restore the ability of T cells to attack tumour cells. Despite these advancements, over 50% of patients experience limited or no benefit from these therapies, highlighting a major challenge in current cancer treatment [6,7]. Recent research has emphasised the significant role of the gut microbiota in influencing cancer development and response to treatment [8]. Although the precise molecular mechanisms underlying these effects are not yet fully understood, emerging research suggests that the composition and function of the gut microbiota influence the anti-tumour immune responses and treatment outcomes of ICI [9,10]. This manuscript explores the complexities of resistance to ICI therapy in melanoma, with a focus on how the microbiota affects tumour biology and the anti-tumour immune response. By examining the impact of the microbiota on ICI efficacy and investigating potential mechanisms for incorporating microbiota-based approaches into clinical treatment, this study aims to provide new insights and identify future research avenues to enhance ICI immunotherapy through microbiota modulation.

## 2. Overview of ICI Therapy in Melanoma

Clinical research on ICI therapy is advancing (Table 1), starting with the most well-known ICMs CTLA-4 and PD-1, where blocking signalling through these molecules has been shown to promote T cell infiltration and expansion at the tumour site and to activate intratumoral natural killer (NK) cells [11,12,13,14]. Since 2011, the U.S. Food and Drug Administration (FDA) has approved several ICIs as first-line treatments for metastatic melanoma, including Ipilimumab, a monoclonal antibody against CTLA-4, and Nivolumab and Pembrolizumab, which block PD-1. These ICIs have become standard first-line treatments for advanced melanoma, demonstrating better response rates than other therapies such as IL-2 and interferons (IFN)-α immunotherapies. Dual targeting of CTLA-4 and PD-1 has been explored in both preclinical and clinical studies. A 2015 phase 3 trial of combined ICI therapy in 945 patients with unresectable stage III and IV melanomas found that those receiving both Ipilimumab and Nivolumab had longer median progression-free survival (11.5 months) compared to those receiving Ipilimumab (2.9 months) or Nivolumab (6.9 months) alone [15]. Another trial reported a higher 3-year overall survival (OS) rate in the combination therapy group (58%) compared to 52% and 34% in the Nivolumab and Ipilimumab groups, respectively [16]. Additionally, the FDA has approved Ipilimumab, Nivolumab, and Pembrolizumab for use as adjuvant therapy in high-risk melanoma, showing improved recurrence-free survival (RFS) and OS compared to high-dose IFN-α2b in resected high-risk melanoma [17]. However, combination therapy was associated with a higher incidence of treatment-related grade 3 or 4 adverse events (59%) compared to 16.3% and 27.3% in patients receiving only Nivolumab and Ipilimumab, respectively [16].

Lymphocyte activation gene 3 (LAG-3) is another checkpoint target that is expressed on activated T cells 3–4 days post-activation, where it suppresses T cell activation and prevents autoimmunity [18,19]. Blocking LAG-3 can restore T cell function and increase tumour infiltration. Research has demonstrated that anti-LAG-3 therapy can improve T cell proliferation, rejuvenate exhausted cytotoxic T lymphocytes, and increase tumour infiltration in murine cancer models [19]. Melanoma patients who responded to the combination therapy of anti-LAG-3 and anti-PD-1 were found to have higher NK cell levels in the tumours and greater transcriptional changes associated with IFN-γ responses, cytotoxicity, and degranulation [20]. Several anti-LAG-3 antibodies (e.g., Relatlimab) are now in clinical trials, both as monotherapies and in combination with other ICI therapies. A clinical study of Relatlimab/Nivolumab combination therapy (Opdualag) reported a significant improvement in progression-free survival (PFS) to 10.2 months versus 4.6 months with Nivolumab alone [21]. At 12 months, PFS was 48.0% for the combination therapy versus 36.9% for Nivolumab monotherapy [21]. Based on these results, the FDA approved this combination therapy in 2022, making LAG-3 the third checkpoint inhibitor approved for cancer treatment [22]. Post-treatment analysis of biospecimens from patients with advanced melanoma demonstrated that dual blockade of LAG-3 and PD-1 leads to enhanced capacity for CD8^+^ T cell receptor signalling and cytotoxicity, despite the retention of an exhaustion profile [23]. However, the increased anti-tumour immune response with combined anti-LAG-3 and anti-PD-1 therapies comes with the cost of higher immune-related adverse events (irAEs), with 21.1% of patients on combination therapy experiencing grade 3 or 4 events compared to 11.1% with anti-PD-1 monotherapy [21]. Thus, optimizing dosages to balance survival benefits and treatment safety remains an active area of research on combined ICI therapies [24,25].

T cell immunoglobulin and mucin domain-containing protein 3 (TIM-3) is a type I transmembrane protein that is upregulated on exhausted NK cells, monocytes, and exhausted CD8 T cells, and its expression was found to be associated with poorer prognoses in melanoma patients [26]. While no anti-TIM-3 antibodies are currently FDA-approved, clinical and pre-clinical trials are underway to evaluate their efficacy in combination with other immune checkpoint inhibitors [27]. Treatment with a soluble TIM-3 blocking antibody has been shown to reverse the exhausted phenotype of NK cells isolated from melanoma patients [26]. Additionally, using a mouse glioma model, dual treatments with anti-TIM-3 antibody and either stereotactic radiosurgery (SRS) or anti-PD-1 were shown to improve survival of the tumour-bearing animals compared with anti-TIM-3 alone; triple therapy also resulted in a significant survival improvement compared to either dual therapy [28]. These results suggest that anti-TIM-3 is most effective as part of a combination therapy aimed at alleviating T cell and NK cell exhaustion.

Moreover, research is also investigating the co-administration of autologous natural killer cells and oncolytic viral adjuvants like TILT-123 or MEM-288 (which is a monoclonal antibody that targets TIM-3) to enhance immune cell activity and improve the anti-tumour response. Clinical studies are also exploring the combination of ICIs with small molecule inhibitors such as Dabrafenib and Trametinib, which target the BRAF-MEK pathway in BRAF-mutated tumours. Furthermore, novel ICI targets are under investigation, including anti-TIGIT (T cell immunoreceptor with Ig and ITIM domains) and anti-ILT4 (Inhibitory Ligand Tool 4) antibodies, which address immune evasion, and anti-CD40 and anti-CD27 antibodies, which aim to stimulate T-cell activation and proliferation. The dual-action ICI M7824, a bifunctional fusion protein targeting both Programmed Death-Ligand 1 (PD-L1) and Transforming Growth Factor-beta (TGF-β), is also being studied to inhibit immune suppression and enhance immune responses [29]. These innovative approaches are being tested across various cancers, including melanoma, with the goals of reducing treatment-related toxicity and improving objective response rates, progression-free survival, and OS. Overall, combination therapies often yield improved response rates but also increase the severity of irAEs, which can affect both OS and PFS [30]. Despite these advances, treatment resistance, which occurs in almost all anti-cancer therapies, remains a major challenge that limits the clinical outcomes of cancers.

**Table 1 ijms-25-10120-t001:** The development of ICI therapies over the last 2 decades.

Clinical Trial	Target	Stage of Melanoma/Study Phase	Key Findings
Ipilimumab (NCT00094653, 2004) [31]	anti-CTLA-4	phase 3 study in advanced melanoma.	Ipilimumab improved OS and provided durable responses in some patients
Nivolumab (NCT00730639, 2008) [32]	anti-PD-1	Phase 1 study in advanced melanoma	Nivolumab demonstrated an acceptable long-term safety profile and durable tumour regression
Pembrolizumab (KEYNOTE-001, NCT01295827, 2011) [33]	anti-PD-1	Phase 1 study in advanced melanoma that progressed after 2 doses of Ipilimumab therapy	Pembrolizumab was well tolerated and demonstrated significant anti-tumour effects in patients previously treated with Ipilimumab
Pembrolizumab (KEYNOTE-002, NCT01704287, 2012) [34]	anti-PD-1	Phase 2 study in advanced melanoma that progressed after Ipilimumab therapy	Pembrolizumab had a higher rate of progression-free survival compared standard-of-care chemotherapy
Nivolumab (CheckMate 037, NCT01721746, 2012) [35]	anti-PD-1	Phase 3 study in advanced melanoma that progressed after Ipilimumab therapy	Nivolumab demonstrated higher objective response rate compared to chemotherapy.
Pembrolizumab, Ipilimumab (KEYNOTE-006, NCT01866319, 2013) [36]	anti-PD-1, anti-CTLA-4	Phase 3 study in advanced melanoma with no more than one prior systemic therapy	Pembrolizumab improved progression-free survival and OS with lower rates of high-grade toxicity compared to Ipilimumab.
Nivolumab, Ipilimumab (CheckMate 064, NCT01783938, 2013) [37]	anti-PD-1, anti-CTLA-4	Phase 2 study in advanced melanoma, without prior treatment or with progression after prior systemic therapy	Sequential treatment with Nivolumab followed by Ipilimumab enhanced efficacy but higher frequency of adverse events, compared to the reverse order.
Nivolumab (CheckMate 066, NCT01721772, 2013) [38]	anti-PD-1	Phase 3 study in advanced melanoma without prior treatment	Nivolumab conferred a significant OS benefit compared to chemotherapy.
Nivolumab, Ipilimumab (CheckMate 067, NCT01844505, 2013) [15].	anti-PD-1, anti-CTLA-4	Phase 3 study in untreated and unresectable advanced melanoma	Nivolumab, with or without Ipilimumab, showed longer progression-free survival and better objective response rates than Ipilimumab alone, especially in PD-L1-negative tumours.
Pembrolizumab, Ipilimumab (KEYNOTE-029, NCT02089685, 2014) [39]	anti-PD-1, anti-CTLA-4	Phases 1 & 2 study in advanced melanoma with no prior ICI therapy	standard-dose Pembrolizumab combined with reduced-dose Ipilimumab offers a manageable toxicity profile and robust anti-tumour activity
Nivolumab, Ipilimumab (CheckMate 238, NCT02388906, 2015) [40]	anti-PD-1, anti-CTLA-4	Phase 3 study as an adjuvant treatment in complete resection of stage IIIB, IIIC, or IV melanoma	Adjuvant Nivolumab significantly improved 12-month recurrence-free survival compared to Ipilimumab and was associated with fewer severe adverse events.
Pembrolizumab (KEYNOTE-054, NCT02362594, 2015) [41]	anti-PD-1	Phase 3 study as an adjuvant treatment in completely resected stage III melanoma with no other prior treatment	Adjuvant Pembrolizumab significantly improved recurrence-free survival compared to placebo, with a manageable safety profile
Ieramilimab, Spartalizumab (NCT02460224, 2015) [42]	anti-PD-1, anti-LAG-3	Phase I/II study in advanced melanoma that progressed after, or were unsuitable for, standard-of-care therapy	Ieramilimab, alone or with spartalizumab, was well tolerated, with modest anti-tumour activity and a safety profile similar to spartalizumab alone.
Pembrolizumab (KEYNOTE-151, NCT02821000, 2016) [43]	anti-PD-1	Phase 1 study in Chinese patients with advanced melanoma that progressed after first-line chemotherapy or targeted therapy	Pembrolizumab therapy was well tolerated in Chinese populations where acral or mucosal subtypes of melanoma are more prevalent and led to durable responses.
Nivolumab, Sotigalimab, Cabiralizumab (NCT03502330, 2018) [44]	anti-PD-1, anti-CD40L, anti-CSF-1R	Phase 1 study in advanced melanoma progressing after prior anti-PD-1/PD-L1 therapy	Combination of sotigalimab and cabiralizumab with or without Nivolumab is well tolerated in patients with anti-PD-1/PD-L1-resistant advanced melanoma.
Nivolumab, Relatlimab (RELATIVITY-047, NCT03470922, 2018) [45]	anti-PD-1, anti-LAG-3	Phases 2 & 3 studies in advanced melanoma with no prior anticancer treatment	Combination of Relatlimab and Nivolumab demonstrated longer progression-free survival than Nivolumab alone.
Nivolumab, Ipilimumab (CheckMate 238, NCT02388906, 2020) [46]	anti-PD-1, anti-CTLA-4	Phase 3 trial in resected stage IIIB-C or IV melanoma	Nivolumab showed a superior recurrence-free survival benefit compared to Ipilimumab, with similar OS rates and a more favourable safety profile
Nivolumab and Ipilimumab (CheckMate 238, 2023) [47]	anti-PD-1, anti-CTLA-4	Resected Stage III/IV Melanoma: 5-Year Efficacy	At 5 years, Nivolumab demonstrated superior recurrence-free survival and distant metastasis-free survival compared to Ipilimumab
Ieramilimab, spartalizumab, 2023 [48]	anti-PD-1, anti-LAG-3	phase 2 study in advanced solid tumours including melanoma, lung cancer, and TNBC with or without receiving prior anti-PD-1/L1 therapy	The combination therapy was well tolerated, showing durable responses, particularly those naive to anti-PD-1/L1 therapy.

## 3. Resistance to ICI Therapy in Melanoma

ICIs can be categorized into two subgroups: primary (innate) resistance and acquired resistance. Primary resistance occurs when patients never respond to treatment, typically due to a lack of tumour-reactive T cell infiltration, which is essential for the effectiveness of ICIs [49]. Acquired resistance, on the other hand, develops after an initial response and is often attributed to intratumoral heterogeneity and tumour diversification [49]. A preclinical study using combined genomic, transcriptomic, and high-dimensional flow cytometric profiling identified several distinct resistance programs involving both tumour-intrinsic and extrinsic factors, leading to multiple resistance mechanisms in melanoma [50].

### 3.1. The Involvement of Tumour-Intrinsic Factors

Several tumour-intrinsic factors are involved in treatment resistance to ICIs, including genetic mutations, altered antigen presentation, signalling pathways, tumour heterogeneity, and expression of other immunosuppressive molecules by the tumour cells. Key factors include mutations in the phosphatase and tensin homolog (PTEN), activation of the WNT/β-catenin signalling pathway, cytokine IFN-γ signalling, loss of heterozygosity in human leukocyte antigen (HLA) genes, and levels of neoantigen expression. PTEN loss or mutation is linked to reduced IFN-γ, granzyme B, and CD8^+^ T cell infiltration [51], while β-catenin signalling activation can result in T cell exclusion in melanoma [52]. WNT/β-catenin signalling activation has also been found to correlate with poor T cell infiltration and a “cold tumour” phenotype. Other contributing factors include dendritic cells (DCs), interleukin-10 (IL-10), transforming growth factor-β (TGF-β), regulatory T cells (Tregs), and reduced CD8^+^ T cell priming and infiltration, all leading to immune evasion and decreased cancer immunosurveillance [53,54]. Additionally, Hugo et al. (2017) identified a set of innate anti-PD-1 resistance signature (IPRES) genes involved in the mesenchymal transition, cell adhesion, extracellular matrix (ECM) remodelling, angiogenesis, and wound healing, which also control the MAPK signalling pathway and inhibit T cell function [55]. This study showed that prior anti-CTLA-4 treatment modifies genomic and transcriptomic features predictive of anti-PD-1 response, highlighting differences in baseline melanoma tumours across immunotherapy trials. Phenotypic switching in melanoma also plays a role in immune evasion, with non-responding tumours showing genes related to undifferentiated and neural crest-like states, while responding tumours display transitory and melanocytic gene signatures [56,57]. Furthermore, mutations in the tumour suppressor p53 (*TP53*) are associated with reduced efficacy of anti-CTLA-4 therapy, possibly due to downregulation of Fas transcription, decreasing susceptibility to CTL-induced apoptosis and reducing anti-CTLA-4 treatment effectiveness [58].

Epigenetic mechanisms, which involve DNA and RNA methylation as well as chromatin remodelling, are intrinsic cellular processes that regulate gene expression without altering the DNA sequence itself [59]. These mechanisms play a crucial role in cancer development and anti-tumour immune responses [60]. Changes in DNA methylation can affect the expression of immune checkpoint genes such as PD-1, PD-L1, PD-L2, and CTLA-4, leading to CD8^+^ T cell exhaustion and altered immune cell recruitment [61]. In melanoma, global hypomethylation is linked to persistent PD-L1 expression and inhibitory cytokine production, contributing to immunosuppression and resistance to ICI therapy [62]. At the molecular level, DNA methyltransferase 3 (DNMT3) mediates new methylation during therapy, while PD-1 promoter demethylation can sustain CD8^+^ T cell exhaustion [63]. DNMT and lysine methyltransferase 6A (KMT6A/EZH2) also suppress Th1-type chemokines like CXCL9 and CXCL10, crucial for T cell recruitment [64]. Global loss of methylation, especially in immunomodulatory genes related to Major Histocompatibility Complex (MHC) and cytokine interactions, correlates with chromosomal instability and reduced anti-tumour immune activity [65]. Chromosomal instability can impact tumour immune infiltration and inflammation via the activation of the cyclic guanosine monophosphate–adenosine monophosphate (GMP–AMP) synthase–stimulator of interferon genes (cGAS–STING) pathway [66], leading to suppression of anti-tumour immunity [67]. STING agonists, which stimulate both innate and adaptive immune responses, represent a novel class of cancer immunotherapy agents [68]. Notably, the expression of STING is epigenetically regulated by histone H3K4 lysine demethylases KDM5B and KDM5C and is activated by opposing H3K4 methyltransferases. Recent research has shown that inhibitors of KDM5 can induce STING expression, suggesting a potential new approach for cancer immunotherapy [68].

Epigenetic dysregulation of transposable elements (TEs) and human endogenous retroviruses (HERVs) can cause inappropriate immune gene activation. Research by Ye et al. highlights that TEs enriched in immune cells are vital for immune regulation [69]. Dysregulation affecting TE-derived enhancers can lead to the inappropriate activation of immune genes during disease progression. HERVs are typically cleared by endogenous immune responses [70], but their reactivation can stimulate immune responses and upregulate viral defence mechanisms, correlating with immune-infiltrating CD8^+^ T cells [71,72]. Conversely, HERV-encoded oncogenes, such as Rec and NP9, can upregulate immunosuppressive pathways like β-catenin, inhibiting immune surveillance [73].

### 3.2. The Involvement of Tumour-Extrinsic Factors

Tumour-extrinsic mechanisms of immune evasion include the influence of immune cells, host microbiota, tumour stroma, and alterations in the local tumour microenvironment (TME), all of which contribute to resistance to ICIs [51,74]. The heterogeneity of immune cell composition in the TME of non-responders, particularly an increased presence of M2 macrophages, is associated with remodelling of the ECM and suppression of the immune system [75]. Even though CD8^+^ T cells may be abundant during ICI therapy, they may lose their cytotoxic effectiveness due to mechanisms such as loss of tumour antigen recognition or tolerance to tumour-associated antigens (TAAs) [76]. Tumour-infiltrating macrophages also contribute to an immunosuppressive TME by secreting factors such as TGF-β, prostaglandins, and IL-10 [77]. These effectors promote the expansion of Treg populations and further recruit and polarize immunosuppressive tumour-associated macrophages (TAMs) [78]. Macrophages also facilitate melanoma progression, as evidenced by their increased density at the invasive front of melanoma lesions [79], and facilitate metastasis through the secretion of metalloproteinases like MMP-9, which remodel the ECM [78]. The ECM, a complex network of secreted molecules, defines tissue architecture and stiffness and influences cell behaviour by supporting cell adhesion, survival, and migration [80]. In melanoma, the ECM becomes highly heterogeneous and dysregulated, incorporating tumour cells, cancer-associated fibroblasts (CAFs), and newly formed blood vessels [80]. ECM molecules also promote TAM infiltration during tumorigenesis and regulate the spatial positioning of TAMs within the TME [81]

Cancer resistance arises from various factors that allow tumour cells to adapt to a changing environment. Therefore, a multifaceted approach may be necessary to address the diverse mechanisms of resistance in treating resistant cancers. Current clinical trials are tackling this issue by incorporating additional factors, such as patient ethnicities and delivery methods, to better evaluate treatment toxicity and efficacy. Moreover, recent studies in the microbiome have identified a significant role of gut or intratumoral microbiota in influencing cancer intrinsic factors, and anti-tumour immune responses, influencing the treatment efficacy and associated irAEs.

## 4. The Role of Microbiota in ICI Therapy

### 4.1. The Association between Microbiota and Cancer

The microbiota that inhabits the human body plays a crucial role in the well-being of its host [82]. Recent research has increasingly focused on its impact on health and disease, revealing that gut microbiota significantly modulates inflammation and immune defence [83,84]. Indeed, intratumoral microbiota can affect therapeutic outcomes by influencing both tumour-intrinsic and extrinsic factors, including genetic and epigenetic expression in tumour cells, signalling pathways like Wnt/β-catenin and NF-κB, metabolism of anti-cancer drugs, and immune cell infiltration and function [85]. Interactions between the host and gut microbes are mostly mediated by the gut metabolome. For example, a study of metabolomic profiles from patients with advanced colorectal adenomas, matched controls, and colorectal cancer (CRC) found that elevated levels of specific bioactive lipids could serve as early indicators of cancer development [86]. Additionally, analysis of the gut microbiome using 16S rRNA gene and shotgun metagenome sequencing revealed that higher microbial community richness and specific microbiome compositions were associated with longer PFS in melanoma patients undergoing immunotherapy [87]. These findings underscore the significant role of gut microbiome interactions in early cancer pathogenesis and therapeutic response.

### 4.2. The Effects of Microbiota on Tumour Genetic and Epigenetic Expression

Research indicates that intratumoral microbiota may contribute to carcinogenesis by releasing toxic byproducts that cause DNA damage, cell cycle arrest, and genomic instability [88]. Gut microbiota can also produce metabolites such as short-chain fatty acids (SCFAs) butyrate and propionate, to modulate host epigenetic machinery. These interactions influence the activity of epigenetic enzymes involved in gene regulation, ultimately affecting immune responses and cancer progression. A study on CRC utilized metagenomic and 16S rRNA gene sequencing to analyse microbiomes in faecal and tissue samples from CRC patients and healthy controls [89]. The study revealed significant differences in DNA methylation patterns between CRC tumours and adjacent normal tissues, with specific microbial-related pathways influencing these changes. Notably, microbial-associated DNA methylation was more evident in adjacent normal tissues but absent in tumours, indicating that gut microbiota and pathogenic bacteria play a crucial role in altering DNA methylation and contributing to CRC development. While the effects of microbiota and their byproducts on epigenetic modifications in melanoma need further investigation, these findings highlight the critical role of host–microbiome interactions in cancer progression.

### 4.3. The Effects of Microbiota in Anti-Tumour Immune Response

The mechanisms by which microbiota enhance anti-tumour immune responses include the upregulation of helper T cell (Th1) and DC functions and the downregulation of regulatory T cells (Tregs), thereby reducing immunosuppression and reinforcing immune activation against tumour cells [90]. By influencing the immune system, microbiota can also affect the efficacy of vaccines and cancer therapies, including ICIs [91,92]. For example, the MIND-DC phase III trial, which randomized 148 stage IIIB/C melanoma patients to receive either autologous DC therapy or a placebo, found that significant baseline differences in gut microbiota and metabolomic profiles affected treatment outcomes [93]. Specifically, the study observed baseline biases in gut microbial composition and metabolic profiles, such as lower levels of *F. prausnitzii* and perturbations in bile acids and acylcarnitines, which were correlated with prognosis and influenced the effectiveness of the dendritic cell therapy [93]. In addition, SCFAs pentanoate and butyrate from microbes can act as enhancers of anti-tumour immunity by modulating cytotoxic T lymphocytes and Chimeric Antigen Receptor (CAR) T cells through metabolic and epigenetic reprogramming, thus improving their efficacy in treating melanoma and pancreatic cancer [94]. The presence of polyclonal neoantigen-specific T cells is also a crucial determinant of effective anti-PD-1 therapy [95]. Interestingly, pre-resection antibiotics administration targeting anaerobic bacteria was found to substantially improve disease-free survival by 25.5% in patients with CRC [96]. Using a mouse CRC model, the authors found that antibiotic treatment generated microbial neoantigens that elicited anti-tumour CD8^+^ T cell response [96]. Gut microbiota can also independently generate epitopes that resemble tumour neoantigens, which can enhance anti-tumour immune responses and contribute to more favourable long-term outcomes with immune checkpoint inhibitors [97].

Microbiota diversity is also a critical determinant of anti-tumour response. Differences in microbiome diversity and composition, along with enhanced systemic anti-tumour immune responses, were observed in melanoma patients who responded to anti-PD-1 ICI compared to non-responders [98]. In a melanoma animal model, the abundance of the commensal microbiota *Bifidobacterium* was associated with increased production of proinflammatory cytokines by DCs, leading to enhanced priming and recruitment of cytotoxic CD8^+^ T cells into the TME [99]. Oral administration of *Bifidobacterium* alone improved tumour control to a degree comparable to that achieved with PD-L1-specific antibody therapy (checkpoint blockade), and combined treatment nearly abolished tumour outgrowth [99]. Similarly, in a preclinical melanoma model, *Lactobacillus reuteri*, found in the gastrointestinal (GI) tract of humans and animals, migrated to the tumour and released the dietary tryptophan catabolite indole-3-aldehyde (I3A). This promoted IFN-γ-producing CD8^+^ T cells and enhanced ICI efficacy [100]. In addition, intratumoral microbiota, particularly the *Lachnoclostridium* genus, positively correlates with CD8^+^ T cell infiltration and chemokine expression and influences survival in cutaneous melanoma [101]. The benefit of microbiota on ICI treatment efficacy is further supported by evidence that faecal microbiota transplantation (FMT) from cancer patients who responded to ICIs improved the anti-tumour effects of PD-1 blockade in germ-free and antibiotic-treated mice [102].

However, intratumoral microbiota can also promote cancer development by inducing genetic mutation, affecting epigenetic modifications, promoting inflammatory response, evading immune destruction, and activating invasion and metastasis. Abnormal gut microbiome composition, associated with dysbiosis, has also been linked to primary resistance to ICIs in patients with advanced cancer [103]. A clinical study analysed microbiomes from 1526 tumours across seven cancer types, including melanoma, and found that each type had a unique microbial profile [104]. Interestingly, intratumor bacteria, mostly intracellular within cancer and immune cells, correlated with tumour characteristics, smoking status, and immunotherapy responses [104]. The mechanisms by which intracellular bacteria within the tumour affect tumour biology and immune responses, and how they differ from extracellular bacteria within the tumour, are currently unknown and require further research. Nevertheless, these findings suggest that modulating the intratumoral microbiome could potentially enhance patient outcomes in immunotherapy.

It is fascinating how gut microbiota can influence anti-tumour immune responses at various sites in the body. Several pathways have been identified through which gut bacteria and their products can reach tumours in different organs. These pathways include passage through the intestinal mucosal barrier into the mesenteric lymph nodes, entry into the systemic circulation, or migration to distant organs due to increased intestinal wall permeability [105]. A recent study using a mouse model of subcutaneous melanoma has uncovered a novel mechanism by which gut microbiota translocation and modulation affect anti-tumour immune responses during ICI therapy [106]. The study demonstrated that ICI treatment induces DC activation, leading to the translocation of gut microbiota to the mesenteric lymph nodes (MLNs) and subsequent remodelling of these nodes. This remodelling facilitates the migration of gut bacteria to tumour-draining lymph nodes (TDLNs) and the subcutaneous primary tumour [106]. This process enhances the activation of effector CD4^+^ and CD8^+^ T cells in both the TDLNs and the tumour, as well as increases leukocyte infiltration and the proportion of IFN-γ and Granzyme B-producing CD8^+^ T cells in the tumour. The presence of MLNs is crucial for the translocation of gut bacteria to extraintestinal tissues. Intestinal DCs are essential for transferring bacteria from the gut to the MLNs but are not required for the subsequent movement from MLNs to TDLNs and the tumour. These findings suggest a complex pathway for microbiota migration beyond the gut, with MLNs serving as a central hub and intestinal DCs acting as transporters to facilitate gut bacterial translocation under specific conditions.

## 5. Predicting Treatment Response to ICI Therapy

Predicting patient responses to ICIs remains a significant challenge in oncology. Several key biomarkers have been explored for this purpose. PD-1 expression on T cells and PD-L1 expression on tumour cells are well-established indicators for predicting responses to anti-PD-1 and anti-PD-L1 therapies. Higher baseline levels of PD-L1 expression generally correlate with improved responses to ICIs [107]. However, PD-L1 expression alone may not always accurately predict outcomes, as its levels can vary across different tumour regions or over time. Increased infiltration of CD8^+^ T cells and other pro-inflammatory immune cells is generally associated with better responses to ICIs [108]. Conversely, high levels of myeloid-derived suppressor cells (MDSCs) correlate with resistance to therapy; reducing these cells enhanced anti-PD-1 therapy in a mouse model of melanoma [109]. Gene expression profiles related to T cell inflammation are also valuable for predicting treatment response. A gene profile associated with T cell inflammation has been connected to improved responses to anti-PD-1 therapy [110]. Variations in expression levels of specific genes, particularly those involved in immune signalling pathways, can help predict patient responses to treatment. Additionally, inflammation-related gene signatures (IRGS) in the TME can distinguish between “cold” and “hot” tumours and predict responses to immunotherapy [111]. Interestingly, High IRGS scores are associated with reduced CD8^+^ T cell infiltration, increased M2 macrophage infiltration, more stroma-activated molecular subtypes, hypoxia, enriched myofibroblast-related signalling, and greater benefit from gemcitabine chemotherapy in prostate cancer [112].

Tumour mutational burden (TMB) is another important predictive factor [107]. The composition and density of immune cell populations within the TME can influence ICI efficacy. In patients with advanced melanoma, high TMB and high inflammatory gene expression scores were associated with longer survival; this was consistent across either Nivolumab and Ipilimumab alone, or in combination [113]. However, TMB alone is not always a reliable predictor of ICI response, as it has been less effective in predicting outcomes for ICIs in various other cancers [114,115]. For instance, ICI responders were found to have more frequent hypomethylation in specific genes, suggesting that methylation profiles could serve as potential predictors of ICI efficacy [116].

Additionally, Microbiota composition has been linked to treatment outcomes. A meta-analysis of four published shotgun metagenomic studies in metastatic melanoma patients identified a faecal microbiome signature associated with responders [117]. The authors found that specific microbial features, such as *Faecalibacterium* and *Barnesiella intestinihominis*, were associated with treatment response. Another meta-analysis study of anti-PD-1 ICI in melanoma patients identified two distinct microbial signatures that differentially affected treatment response [118]. Patients enriched for Actinobacteria and the *Lachnospiraceae/Ruminococcaceae* families of *Firmicutes* showed favourable clinical responses. In contrast, Gram-negative bacteria such as *Streptococcaceae spp* were associated with an inflammatory gene signature, increased blood neutrophil-to-lymphocyte ratio, and poor outcomes with distinct immune-related adverse effects [118]. Moreover, longitudinal microbiome profiling by Weersma et al. revealed several microbial species-level genome bins (SGBs) and pathways that could differentiate patients who achieved PFS of 12 months or longer from those with shorter PFS after ICI treatment [119].

Taken together, predicting responses to ICI therapy involves a multifaceted approach that includes tumour-intrinsic factors, immune cell infiltration, epigenetic profiles, and microbiota signatures. By leveraging these predictive biomarkers, clinicians can better tailor treatment strategies and improve outcomes for melanoma patients undergoing ICI therapy.

## 6. Summary and Future Perspective

In conclusion, the development of ICIs is advancing with a focus on identifying new targets and optimizing combination therapies tailored to individual patients. This progress involves balancing the therapeutic efficacy of ICIs with the management of irAEs [120]. Emerging research on the microbiome underscores its potential role in influencing cancer epigenetics, immune responses, and ICI outcomes (see Figure 1). The concept of a “symbiotic microecosystem” within tumours comprising intratumoral microbiota, tumour cells, and immune cells highlights a fundamental interplay in cancer development and treatment success [121]. Recent findings indicate that specific microbiota profiles can enhance responses to PD-1 inhibitors, suggesting promising interventions such as FMT, probiotics, and dietary adjustments [8,122]. These strategies could improve the presence of anti-tumour microbiota and, consequently, treatment outcomes. In addition, further research is needed to elucidate their role in tumour progression, identify strategies to promote anti-tumour microbiota infiltration into the tumours, and the molecular mechanisms behind these processes. Taken together, the field of anti-cancer therapy is evolving rapidly, with significant promise for both ICI and microbiome-based treatments. These approaches potentially offer more cost-effective and efficient strategies to improve cancer treatment outcomes in the future.

## Figures and Tables

**Figure 1 ijms-25-10120-f001:**
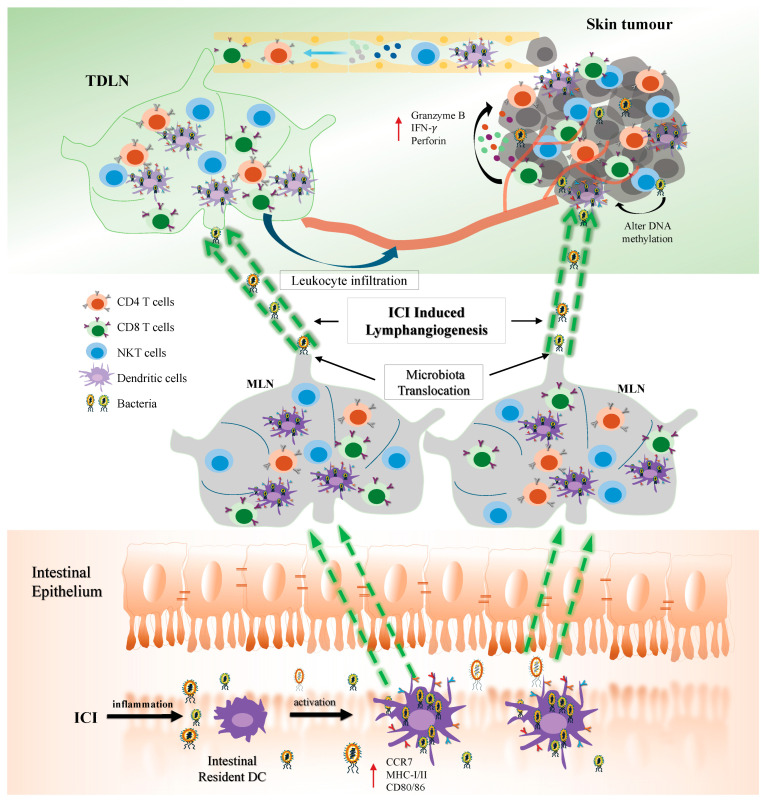
Synergistic response of ICI and gut microbiota. Inflammation-induced by ICI stimulates gut microbiota uptake by and activation of intestinal DCs, accompanied by the upregulation of MHC, CD86, and CCR7. This process facilitates the translocation of bacterial-retaining DCs into the MLNs, leading to bacterial translocation to the TDLNs and the tumour, where they alter epigenetic expression of the tumour cells, and upregulate the production of pro-inflammatory cytokines by effector immune cells, such as CD8 T cells and NKT cells. This procedure also results in increased leukocyte infiltration into the tumour and secretion of Granzyme B and perforin to initiate tumour elimination.

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
