# Peer review of "Immune Checkpoint Inhibitor Therapy for Metastatic Melanoma: What Should We Focus on to Improve the Clinical Outcomes?"

_ijms, 2024, doi:10.3390/ijms251810120_

Round 1
Reviewer 1 Report
Comments and Suggestions for Authors
The manuscript reviews various aspects of ICB therapy and considers how it could be improved. Overall the review is mostly redundant of many other general reviews of ICB and its problems and opportunities but since it is relatively accurate that is not a big problem. The most interesting sections that have not been already extensively reviewed is the impact of intratumoral microbiome, a topic that could be the basis of the entire review. Unfortunately that section is short and not the focus of the review. One improvement for the microbiota section is that the authors move quickly between intratumoral and gut microbiota and summarize various reported studies but do not make clear for each what anatomic location had the reported microbes.
The writing is generally acceptable but has numerous phrasing problems such as line 66 “to prevent access inflammation”, likely access should be excess; line 70 “its efficacy be improved”; line 90 “(10 months %)”. Line 390 “It is interesting how the gut microbiota reaches the skill and elicit anti-tumour immune response” is not an understandable sentence. This list is not complete but careful editing by authors should solve most of the phrasing issues.
Line 333 discusses PD-L1 expression effects on T cells but does not note what cells are expressing PD-L1 and what PD-L1 is signaling through either externally or internally.
Comments on the Quality of English Languageas noted, some typos and unclear or grammatically incorrect phrasing but overall the English is fine
Author Response
Comments 1: The manuscript reviews various aspects of ICB therapy and considers how it could be improved. Overall the review is mostly redundant of many other general reviews of ICB and its problems and opportunities but since it is relatively accurate that is not a big problem. The most interesting section that has not been already extensively reviewed is the impact of the intratumoral microbiome, a topic that could be the basis of the entire review. Unfortunately, that section is short and not the focus of the review. One improvement for the microbiota section is that the authors move quickly between intratumoral and gut microbiota and summarize various reported studies but do not make clear for each what anatomic location had the reported microbes.
Response 1: We appreciate the reviewer’s feedback. We acknowledge that the ICB therapy content is well-covered in existing literature. We agree that the microbiome's role is a key area of interest and have significantly revised the manuscript to highlight this topic more prominently. We have reorganized the microbiome section and added more detailed information to enhance its focus and clarity. Additionally, we have clarified the distinction between intratumoral and gut microbiota, specifying the anatomical locations of the microbes discussed.
Comments 2: The writing is generally acceptable but has numerous phrasing problems such as line 66 “to prevent access inflammation”, likely access should be excess; line 70 “its efficacy be improved”; line 90 “(10 months %)”. Line 390 “It is interesting how the gut microbiota reaches the skill and elicit anti-tumour immune response” is not an understandable sentence. This list is not complete but careful editing by authors should solve most of the phrasing issues.
Response 2: Thank you for highlighting these issues. We have corrected the typos and phrasing errors throughout the manuscript, including those mentioned. The manuscript has undergone careful editing to improve the clarity and accuracy of the language.
Comments 3: Line 333 discusses PD-L1 expression effects on T cells but does not note what cells are expressing PD-L1 and what PD-L1 is signalling through either externally or internally.
Response 3: We appreciate this observation. The relevant sentence has been removed as part of a broader content rearrangement and refocusing of the manuscript. We have adjusted the content to better address the points raised.
Reviewer 2 Report
Comments and Suggestions for Authors
In this review, M. Hossain et al. have summarized the findings related to immune checkpoint inhibitor therapy and factors that could influence the outcome of this treatment in melanoma. The manuscript reaches a decent quality.
There are some minor issues:
1. Please define the acronyms/Abbreviations the first time they appear: IL (line 31), FDA (line 33), IFN (43), APC (line 78), TCR (line 81), TILs (119), MHC (line 141), NK (line 145), MAPK/Erk and PI3K/Akt (line 148), mo (line 178), PTEN (line 225), HLA (line 227), TME (228), TGF-β (line 230), Treg (line 231), ECM (239), VEGF (241), DNMT3 (281), KMT6A (287), GMP-AMP (303), TAMs (338), and GI (387).
2. Please correct and rephrase the statement “Recombinant IFN-⍺-the FDA approved 2b” in line 43; please correct LAG-3d (line 174) and CD8+ Td (line 188).
3. Lines 112-114: Please rephrase the statement: “An overall response rate (ORR) of 40% to 50% and 5-year objective survival (OS) rates of 30 to 40% in patients with metastatic melanoma receiving Nivolumab and Pembrolizumab ICI”.
4. Please choose an abbreviation and use it in the same way throughout the manuscript: IFN-⍺ or IFN-a (line 51), irAEs (line 112) or IRAEs (lines 180, 447), overall survival (OS) in line 111 or objective survival (OS) in line 113.
5. I suggest to use either upper- or lower- case letters for names of drugs (nivolumab, ipilimumab, pembrolizumab, and relatlimab).
6. In the paragraph in lines 164-182 is a mixture of data from references 41 and 43. Reference 43 (Long et al, 2023) reported updated data from ClinicalTrials.gov number, NCT03470922. I suggest you use the data from reference 43 (Long et al, 2023):
- replace the reference 41 with reference 43 in lines 169 and 172;
- “PFS rates at 12 months were 48.0% versus 36.9%”, please replace the data in lines 170-172;
- “Grade 3/4 treatment-related adverse events were observed in 21.1% of patients treated with nivolumab + relatlimab versus 11.1% treated with nivolumab”, please replace the data in lines 180-181.
7. Please add the Author Contributions and Conflicts of Interest.
8) A figure/cartoon summarizing the information discussed in the review would be very helpful for future readers. Also, a table summarizing studies to analyze the impact of ICIs in melanoma would be useful.
Author Response
Comment 1: Please define the acronyms/Abbreviations the first time they appear: IL (line 31), FDA (line 33), IFN (43), APC (line 78), TCR (line 81), TILs (119), MHC (line 141), NK (line 145),MAPK/Erk and PI3K/Akt (line 148), mo (line 178), PTEN (line 225), HLA (line 227), TME (228), TGF-β (line 230),Treg (line 231), ECM (239), VEGF (241), DNMT3 (281), KMT6A (287), GMP-AMP (303), TAMs (338), and GI (387).
Response 1: Thank you for pointing this out. We have revised the manuscript to define all acronyms and abbreviations the first time they appear, ensuring clarity and consistency throughout the text.
Comment 2: Please correct and rephrase the statement “Recombinant IFN-⍺-the FDA approved 2b” in line 43; please correct LAG-3d (line 174) and CD8+ Td (line 188).
Response 2: We have corrected the typos in line 43 to “Recombinant IFN-⍺, FDA approved (2b).” Additionally, we have corrected “LAG-3d” to “LAG-3” and “CD8+ Td” to “CD8+ T cells” in the respective lines.
Comment 3: Lines 112-114: Please rephrase the statement: “An overall response rate (ORR) of 40% to 50% and 5-year objective survival (OS) rates of 30 to 40% in patients with metastatic melanoma receiving nivolumab and pembrolizumab ICI”.
Response 3: This statement was deleted during the content re-arrangement. As a result, it is no longer present in the manuscript.
Comment 4: Please choose an abbreviation and use it in the same way throughout the manuscript: IFN-⍺ or IFN-a (line 51), irAEs (line 112) or IRAEs (lines 180, 447), overall survival (OS) in line 111 or objective survival (OS) in line 113.
Response 4: We have standardized the use of abbreviations throughout the manuscript. We now consistently use “IFN-⍺,” “irAEs,” and “overall survival (OS)” in all instances.
Comment 5: I suggest to use either upper- or lower- case letters for names of drugs (nivolumab, ipilimumab, pembrolizumab, and relatlimab).
Response 5: We have revised the manuscript to consistently use upper-case for drug names: Nivolumab, Ipilimumab, Pembrolizumab, and Relatlimab.
Comment 6: In the paragraph in lines 164-182 is a mixture of data from references 41 and 43. Reference 43 (Long et al, 2023) reported updated data from ClinicalTrials.gov number, NCT03470922. I suggest you use the data from reference 43 (Long et al, 2023):
- replace the reference 41 with reference 43 in lines 169 and 172;
- “PFS rates at 12 months were 48.0% versus 36.9%”, please replace the data in lines 170-172;
- “Grade 3/4 treatment-related adverse events were observed in 21.1% of patients treated with nivolumab + relatlimab versus 11.1% treated with nivolumab”, please replace the data in lines 180-181.
Response 6: We have updated the manuscript according to your suggestions. Reference 43 has been substituted for reference 41 in lines 164-182. The data in lines 170-172 and 180-181 have been corrected to reflect the updated information
Comment 7: Please add the Author Contributions and Conflicts of Interest.
Response 7: We have added sections detailing Author Contributions and Conflicts of Interest to the manuscript to meet the required standards.
Comment 8: A figure/cartoon summarizing the information discussed in the review would be very helpful for future readers. Also, a table summarizing studies to analyze the impact of ICIs in melanoma would be useful.
Response 8: We have included a figure summarizing the key findings of the review and a table that summarizes the studies analyzing the impact of immune checkpoint inhibitors (ICIs) in melanoma. We believe these additions will enhance the manuscript’s clarity and utility for readers.
Thank you again for your thorough review and constructive feedback.